# Peroxisome Proliferator-Activated Receptor α/δ/γ Activation Profile by Endogenous Long-Chain Fatty Acids

**DOI:** 10.3390/ijms262412020

**Published:** 2025-12-13

**Authors:** Akihiro Honda, Aoi Hosoda, Waka Kamichatani, Midori Ogasawara, Shiori Miyazaki, Nonoka Kashiwagi, Shotaro Kamata, Isao Ishii

**Affiliations:** Department of Health Chemistry, Showa Pharmaceutical University, Machida, Tokyo 194-8543, Japan

**Keywords:** coactivator recruitment, 15-deoxy-Δ^12,14^-prostaglandin J2, endogenous ligand, ligand-binding domain, non-esterified fatty acids, nuclear receptor, peroxisome proliferator-activated receptor, therapeutics, time-resolved fluorescence resonance energy transfer, transcriptional factor

## Abstract

There is a wealth of information available about endogenous fatty acid ligands for peroxisome proliferator-activated receptor α/δ/γ (PPARα/δ/γ); however, there are few comparative studies of PPARα/δ/γ activation using standardized experimental systems. This study investigated which of 14 major free long-chain fatty acids (LCFAs: C12:0–C22:6) and 15-deoxy-Δ^12,14^-prostaglandin J2 (15d-PGJ2) activate PPARα/δ/γ using a coactivator recruitment assay. We recently discovered that eight different synthetic PPAR agonists recruit four different coactivator peptides (PGC1α, CBP, SRC1, TRAP220) with varying potency and efficacy, so we examined the ligand-concentration-dependent recruitment of these four coactivators. All 15 fatty acids (FAs) activated PPARα/δ at high concentrations, but only palmitic acid, stearic acid, oleic acid, and linoleic acid significantly activated PPARα/δ at physiologically relevant concentrations. Lauric acid, myristic acid, palmitic acid, and 15d-PGJ2 activated PPARγ at high concentrations, but only palmitic acid slightly activated PPARγ at physiologically relevant concentrations. FA ligands exhibited different coactivator preference compared to synthetic PPAR agonists, including approved drugs such as pemafibrate, seladelpar, and pioglitazone, suggesting that these agonists may regulate target gene transcription in a different manner than natural FA ligands. Such differences may be relevant to the pathogenesis of side effects of synthetic PPAR agonists occasionally observed in (pre)clinical studies.

## 1. Introduction

Peroxisome proliferator-activated receptors (PPARs) are ligand-activated transcription factors belonging to the nuclear receptor family that regulate lipid and glucose metabolism. They consist of α/δ/γ subtypes, which exhibit diverse tissue distribution and functions in mammals [1,2]. Several PPAR agonists were/are used as medications (e.g., the triglyceride-lowering fibrates and the antidiabetic pioglitazone) or are being developed as next-generation therapeutics for metabolic diseases such as type 2 diabetes, dyslipidemia, and metabolic dysfunction-associated fatty liver disease, chronic inflammatory disease, cardiovascular disease, autoimmune disease, neurodegenerative disease, and cancer [2,3,4,5,6]. There is a wealth of information available about the endogenous ligands of PPARs, which are presumably free long-chain fatty acids (LCFAs) produced prolifically via the lipolysis of dietary or stored triacylglycerols or certain types of eicosanoids derived from membrane phospholipids [2,7]. However, there are limited examples of comparisons using standardized experimental systems, and eicosanoids are produced in such small amounts that they are difficult to quantify (or even identify), casting doubt on their involvement at physiological concentrations [8,9]. Previous studies have proposed several endogenous fatty acid (FA) ligands for PPARα/δ/γ, primarily based on (1) cell-free binding assays using radioactive ligands, such as titration assays [10,11] and scintillation proximity assays [12,13]; (2) cell-based PPAR-ligand-binding domain (LBD)–peroxisome proliferator response element-mediated transactivation reporter gene assays [11,14]; and (3) cell-based assays for detecting the induction of several cellular responses inhibited by PPAR antagonists [15] or PPAR mRNA silencing/gene deletion [11,16]. However, PPARα/δ/γ binding does not directly indicate activation because antagonists or even inverse agonists can be substituted for radioligands. In cell-based assays, externally sourced LCFAs may not penetrate the plasma membrane, enter the nucleus, bind directly (without metabolic pathways), and activate PPARα/δ/γ at physiological (in situ) concentrations.

This study investigated the activation of human PPARα/δ/γ by 14 types of LCFAs (C12:0–C22:6), which are relatively abundant in the body, and 15-deoxy-Δ^12,14^-prostaglandin J2 (15d-PGJ2), an eicosanoid that is present in only trace amounts but is considered to be an endogenous PPARγ ligand, although its role is still subject to debate [8,9]. We utilized a cell-free time-resolved fluorescence resonance energy transfer (TR-FRET) assay for high-sensitivity detection of ligand-induced coactivator recruitment to PPARα/δ/γ-LBD [17,18,19,20]. Agonist binding induces a conformational change in PPARα/δ/γ-LBD, which in turn recruits the coactivator complex to promote transcriptional activation of specific target genes [21,22]. We recently discovered that eight PPAR agonists recruit four different coactivator peptides (PPARγ coactivator-1α [PGC1α], cyclic AMP-responsive element-binding protein-binding protein [CBP], steroid receptor coactivator 1 [SRC1], and thyroid hormone receptor-associated protein 220 [TRAP]) with varying potency and efficiency [19,20]. Therefore, this study compared the activity of these 15 types of naturally occurring FAs in recruiting these coactivator peptides to PPARα/δ/γ-LBD using the simplest standardized system. Ultimately, we found that each PPAR prefers different FAs as ligands and that approved PPAR drugs (pemafibrate, seladelpar, and pioglitazone [6,23]) alter coactivator preferences from FA ligands.

## 2. Results

First, we examined the recruitment of four coactivators to the PPARα/δ/γ-LBD by synthetic Glaxo–Wellcome (GW) series full agonists and approved PPARα/δ/γ-selective agonists: pemafibrate, seladelpar (approved for the treatment of primary biliary cholangitis in August 2024), and pioglitazone (Figure 1). As described later in the Materials and Methods Section, we have improved the TR-FRET assay used in our previous paper [19] to be more sensitive, faster, and less expensive. The maximal responses induced by PPARα-selective GW7647 (Figure 1A), PPARδ-selective GW501516 (Figure 1B), and PPARγ-selective GW1929 (Figure 1C) differed among the four coactivators: PGC1α > SRC1 > CBP > TRAP in PPARα; CBP > PGC1α > TRAP > SRC1 in PPARδ; and CBP > PGC1α ≈ TRAP > SRC1 in PPARγ. The maximal responses induced by 1 µM of those GW agonists were considered as 100% of the responses in later experiments. Pemafibrate induced full activation of PPARα in the recruitment of CBP, TRAP, and SRC1 but only 80% activation in the recruitment of PGC1α (Figure 1D). Seladelpar induced full activation of PPARδ in the recruitment of TRAP and 60–80% activation in the recruitment of PGC1α, CBP, and SRC1 (Figure 1E), whereas pioglitazone acted as a partial PPARγ agonist for all four coactivators at varying activation levels: CBP > TRAP > PGC1α > SRC1 (Figure 1F).

Next, we examined the recruitment of four coactivators to the PPARα/δ/γ-LBD by 14 major LCFAs (C12:0–C22:6), including saturated and unsaturated FAs, the latter of which included *n*–3 or *n*–6 essential FAs, and 15d-PGJ2 (Figure 2, Figure 3 and Figure 4). The blood concentrations of the LCFAs varied greatly [24], depending significantly on diet. In our previous study involving young male volunteers [18], the baseline levels of serum palmitic acid, stearic acid, oleic acid, and linoleic acid (114, 48, 133, and 78 µM, respectively) measured after a 12 h fasting period decreased to 41, 21, 17, and 7 µM, respectively, 2 h after meal intake. To establish a physiologically relevant concentration range, we used the mean plasma levels after overnight fasting from three healthy cohorts [24,25,26] as a reference: the mean plasma concentrations of 28 individuals (average age: 56.4 ± 18.9 years; 0.68 µM for C12:0; 12.63 µM for C14:0; 66.44 µM for C16:0; 18.92 µM for C18:0; 0.21 µM for C20:0; 10.08 µM for C16:1; 38.91 µM for C18:1; 136.29 µM for C18:2; 1.16 µM for C18:3 including both α- and γ-linolenic acid; 1.18 µM for C20:4; and 0.67 µM for C20:5 [24]), a total of 22 L plasma from 100 individuals (40–50 years; 0.542 µM for C20:3 and 0.990 µM for C22:6 [25]), and 44 individuals (average age: 38.0 ± 10.4 years; 0.144 nM for 15d-PGJ2 [26]). Single-digit ranges centered on the mean concentrations of human plasma samples are shown in the light blue fields in Figure 2, Figure 3 and Figure 4.

All 15 FAs recruited all four coactivators to PPARα-LBD to varying degrees, and the general coactivator preference of PPARα was CBP > TRAP ≈ SRC1 ≈ PGC1α (Figure 2). In particular, palmitic acid (C16:0), stearic acid (C18:0), oleic acid (C18:1), and linoleic acid (C18:2), which are abundant in human plasma, markedly activated PPARα (38.7–64.1% of the maximal response for CBP recruitment) at physiologically relevant concentrations (Figure 2C,D,G,H). Notably, lauric acid (C12:0) and all unsaturated FAs displayed bell-shaped responses (Figure 2A,F–O). Similarly, all 15 FAs recruited all four coactivators to PPARδ-LBD with general coactivator preference of PGC1α > TRAP > CBP ≈ SRC1, but only palmitic acid, stearic acid, and oleic acid markedly activated PPARδ (21.3–97.3% of the maximal response for PGC1α recruitment) at physiologically relevant concentrations (Figure 3). In particular, palmitic acid behaved like a PPARδ full agonist (Figure 3C). Most FAs, except for palmitic acid and stearic acid, displayed bell-shaped responses (Figure 3A,B,E–O). On the other hand, activation of PPARγ by FAs was limited (Figure 4). Only palmitic acid slightly induced the recruitment of four types of coactivators at physiologically relevant concentrations (300 µM), ranging from 3.2% to 25.5% of the maximum response and varying considerably depending on the coactivator (Figure 4C). Furthermore, lauric acid, myristic acid, and 15d-PGJ2 activated PPARγ only at supraphysiological concentrations (Figure 4A,B,O). The general coactivator preference of PPARγ was PGC1α > CBP > TRAP > SRC1 for lauric acid, myristic acid, and palmitic acid (Figure 4A–C) and CBP > PGC1α > TRAP > SRC1 for 15d-PGJ2 (Figure 4O).

## 3. Discussion

LCFAs (FAs with ≥12 carbons) and medium-chain FAs (6–11 carbons) are mostly derived from dietary or stored triacylglycerols, whereas short-chain FAs (<6 carbons) are produced by gut microbial fermentation of dietary fiber. All are important energy sources [27]. Humans lack Δ^15^- and Δ^12^-desaturase and, consequently, *n*–3 (α-linolenic acid, EPA, and DHA) and *n*–6 (linoleic, γ-linolenic, dihomo-γ-linolenic, and arachidonic acid) FAs can only be obtained through diet. 15d-PGJ2 is produced via dehydration from prostaglandin D2, a principal cyclooxygenase-2 product during the inflammatory processes. Lipid-binding protein members, including free fatty acid receptors (FFARs) and prostanoid receptors in the plasma membrane, fatty acid-binding proteins (FABPs) in the cytoplasm, and PPARs in the nucleus, are functional sensors of such free FAs [2,27]. The ligand-binding pockets of PPARα/δ/γ, which have volumes of 1400, 1300, and 1600 (or 1440) Å^3^, respectively, are larger than those of other nuclear receptors (600–1100 Å^3^) and could thus contain several ligand molecules simultaneously [2,18,28].

PPARα is pivotal in the adaptive response to fasting. During fasting, circulating free FAs produced by lipolysis and β-oxidation activate PPARα and induce transcription of its responsive gene, including *PPARA* [29,30]. PPARα was activated by all 15 FAs tested at physiologically relevant or supraphysiological concentrations (Figure 2). We previously found that palmitic acid, stearic acid, oleic acid, and arachidonic acid bind similarly to the Center and Arm II regions of PPARα-LBD on X-ray crystallographic analyses of cocrystals, although palmitic acid and stearic acid—but not oleic acid or arachidonic acid—activated PPARα at 1 mM concentration in the coactivator recruitment [18]. The present study has thus elucidated the reason. Excessive intake of essential fatty acids through supplements (mainly widely sold EPA/DHA products) can lead to elevated blood levels, potentially resulting in unintended activation of PPARα.

Although PPARδ is essential in mouse embryonic development [29], its physiological roles and those of its endogenous ligands are largely unknown [31]. However, palmitic acid, arachidonic acid, leukotrienes, lipoxins, and 5-hydroxyeicosatetraenoic acids (5-HETEs) are considered as PPARδ ligand candidates [2]. The present study has identified stearic acid and oleic acid as part of the list (Figure 3D,G). Activation of PPARδ promotes FA oxidation, which in turn induces the activation of PPARα, and activation of PPARα also promotes FA oxidation in hepatocytes. Therefore, these aforementioned LCFAs may be involved in such synergistic effects [32].

Finally, although PPARγ was highly activated by GW1929 (Figure 1C) and pioglitazone (Figure 1F), it was not activated by most LCFAs except lauric acid, myristic acid, palmitic acid, and 15d-PGJ2 (Figure 4). Linoleic acid, arachidonic acid, EPA, 9-/13-hydroxyoctadecadienoic acid (9-/13-HODE), and 15d-PGJ2 were previously considered as natural ligands for PPARγ [2,28,32]. However, the present study identified palmitic acid rather than linoleic acid, arachidonic acid, EPA, and 15d-PGJ2 as possible natural ligands for PPARγ (Figure 4C). 15d-PGJ2 significantly activated PPARγ at concentrations exceeding micromolar levels (Figure 4O). This concentration is more than three orders of magnitude higher than the concentrations found in the plasma of healthy individuals or patients with strokes, diabetes, or mania (12–344 pM) [26,33,34] or the 3T3-L1 fibroblast cells/culture medium (~5 pM/~1 nM) [9]. While complete loss of PPARγ activity is (embryonic) lethal, as observed in PPARγ-knockout mice [29], it may remain largely inactive under physiological conditions. This inherent low activity may explain why various PPARγ agonists often cause serious side effects in clinical studies, leaving only safer partial agonists, such as pioglitazone (Figure 1F), on the market. Still, some PPARγ agonists may have off-target effects, such as PPAR-independent genomic actions or non-genomic actions controlled by post-translational modifications [35,36,37], or may have entirely novel activities in combination with imatinib (a BCR-ABL tyrosine kinase inhibitor) or MEK inhibitors in cancer treatment [38,39,40]. Clinical trials using multiple PPAR agonists are being conducted for autoimmune diseases, inflammatory diseases, infectious diseases, and malignant tumors [3]. Under these inflammatory conditions, the blood and local concentrations of many FAs are known to increase or decrease, and it is possible that these endogenous FAs may be involved in the exacerbation or remission of these diseases via PPARs.

In conclusion, palmitic acid appears to be a potential endogenous PPAR pan-agonist, and stearic acid and oleic acid may be endogenous PPARα/δ dual-agonists. Furthermore, the approved drugs, pemafibrate, seladelpar, and pioglitazone, alter coactivator preferences of FA ligands and may thus regulate target gene transcription differently from natural FA ligands.

## 4. Materials and Methods

### 4.1. Expression and Purification of Recombinant PPARα/δ/γ-LBD

Human PPARα-LBD (amino acids [AAs] 200–468), PPARδ-LBD (AAs 170–441), and PPARγ-LBD (isoform 1; AAs 203–477) proteins were expressed as 6× His-tagged proteins in Rosetta (DE3)pLysS competent cells (Novagen [Merck KGaA], Darmstadt, Germany) using the pET28a vector (Novagen). The proteins were then purified using a cobalt-based metal affinity column (TALON Metal Affinity Resin; Takara Bio, Shiga, Japan), a HiTrap Q anion exchange column (GE Healthcare, Chicago, IL, USA), and a HiLoad 16/600 Superdex 75-pg gel filtration column (GE Healthcare), as described previously [18,41]. Subsequent processing revealed that 80% of PPARα-LBD recombinant proteins contained an endogenous FA (contaminants from E. coli during preparation), which were delipidized (fatty acid removal) with ethanol as described previously [18,41].

### 4.2. Coactivator Recruitment TR-FRET Assay

The activation state of each PPARα/δ/γ subtype was determined using a TR-FRET assay [19,20]. This assay detects the physical interaction between recombinant PPARα/δ/γ-LBD proteins and biotin-labeled coactivator peptides containing consensus α-helical Leu-X-X-Leu-Leu (LXXLL, X: any amino acid) motifs. The four coactivator peptides used were PGC1α AAs 137–155 (EAEEPSLLKKLLLAPANTQ), CBP AAs 56–80 (SGNLVPDAASKHKQLSELLRGGSGS), SRC1 AAs 676–700 (CPSSHSSLTERHKILHRLLQEGSPS), and TRAP220 AAs 631–655 (PVSSMAGNTKNHPMLMNLLKDNPAQ), all of which were synthesized by GenScript (Tokyo, Japan). A 9.5 µL aliquot of PPARα/δ/γ-LBD (400 nM in Buffer A: 20 mM HEPES-NaOH [pH 7.4], 150 mM NaCl, 1 mM EDTA, 0.005% [*v*/*v*] Tween 20, 0.2% FA-free bovine serum albumin), 0.5 µL of 100× ligand solution (in ethanol for FAs and dimethyl sulfoxide for synthetic ligands), and 5 µL of biotin-labeled coactivator peptide (1 µM in Buffer A) were added to a single well of a 384-well low-volume, white round-bottom, polystyrene non-binding surface microplate (No. 4513, Corning, Charlotte, NC, USA). Next, a 5 µL aliquot of 4 nM Eu-W1024-labeled anti-6×His antibody/40 nM ULight-streptavidin (PerkinElmer, Shelton, CT, USA) was added to each well, and the microplate was incubated in the dark at room temperature for 2 h. FRET signals were measured using an Infinite^®^ 200 Pro F plex microplate reader (Tecan, Männedorf, Zürich, Switzerland) with a single excitation filter (320/25) and two emission filters (620/10 and 665/9). The measurement parameters were 200 s integration time and 100 µs delay time. Emission at 665 nm was due to ULight-FRET, while that at 620 nm was due to Eu-W1024. The 665/620 ratio was calculated and normalized to the negative control reaction using 1% dimethyl sulfoxide. Coactivator recruitment was expressed as a percentage of the maximal responses induced by 1 µM synthetic PPARα/δ/γ full agonists (GW7647, GW501516, and GW1929 for PPARα/δ/γ, respectively).

The improvements over the similar TR-FRET assay reported in our previous paper [19] were (1) the use of a new TECAN microplate reader equipped with appropriate filters, which offers 10 times more detection sensitivity and less than 1/10 the measurement time; (2) the reduction in the amounts of expensive Eu- and ULight-labeled reagents used by half; (3) the delipidation of PPARα-LBD (see above); and 4) the removal of 1 mM dithiothreitol from the assay mixture (not for this experiment, but for future experiments). GW7647, GW501516, GW1929, pioglitazone, linoleic acid (C18:2), α-linolenic acid (C18:3 [α]), eicosapentaenoic acid (C20:5; EPA), and docosahexaenoic acid (C22:6; DHA) were purchased from Cayman Chemical (Ann Arbor, MI, USA). Pemafibrate was kindly provided by Kowa Co., Ltd. (Tokyo, Japan). Seladelpar was purchased from ChemScene (Monmouth Junction, NJ, USA). Arachidic acid (C20:0), γ-linolenic acid (C18:3 [γ]), dihomo-γ-linolenic acid (C20:3), and arachidonic acid (C20:4) were purchased from Tokyo Chemical Industry (Tokyo, Japan). Lauric acid (C12:0), myristic acid (C14:0), palmitoleic acid (C16:1), and oleic acid (C18:1) were purchased from Nacalai Tesque (Tokyo, Japan). Palmitic acid (C16:0), stearic acid (C18:0), and 15d-PGJ2 were purchased from MilliporeSigma (Burlington, MA, USA).

## Figures and Tables

**Figure 1 ijms-26-12020-f001:**
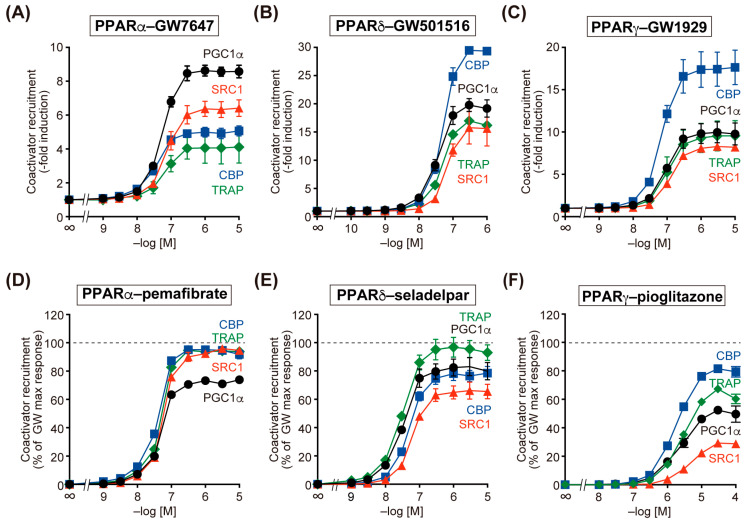
PPARα/δ/γ activation by synthetic full agonists and clinically approved drugs in a coactivator recruitment assay. Agonist-induced recruitment of four coactivator peptides (PGC1α in black circles; CBP in blue squares; SRC1 in red triangles; and TRAP in green diamonds) to human PPARα/δ/γ-LBDs was examined by using a cell-free TR-FRET assay. Concentration-dependent activity of synthetic PPARα/δ/γ full agonists ((**A**) GW7647; (**B**) GW501516; and (**C**) GW1929) and approved drugs (**D**) pemafibrate, (**E**) seladelpar, and (**F**) pioglitazone, respectively). The maximal responses induced by 1 µM GW compounds in (**A**–**C**) were used as the 100% responses in (**D**–**F**). Data are expressed as the mean ± standard error (S.E.) of three independent experiments with duplicate samples. All raw data are available in Appendix A.

**Figure 2 ijms-26-12020-f002:**
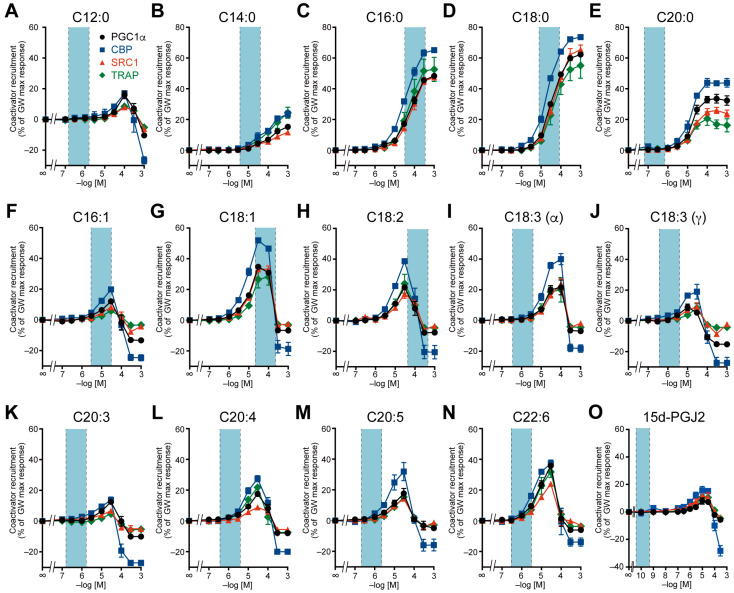
PPARα activation by 15 endogenous fatty acids in a coactivator recruitment assay. Agonist-induced recruitment of four coactivator peptides (PGC1α in black circles; CBP in blue squares; SRC1 in red triangles; and TRAP in green diamonds) to human PPARα-LBD was examined using a cell-free TR-FRET assay. Concentration-dependent activity of (**A**) lauric acid, (**B**) myristic acid, (**C**) palmitic acid, (**D**) stearic acid, (**E**) arachidic acid, (**F**) palmitoleic acid, (**G**) oleic acid, (**H**) linoleic acid, (**I**) α-linolenic acid, (**J**) γ-linolenic acid, (**K**) dihomo-γ-linolenic acid, (**L**) arachidonic acid, (**M**) EPA, (**N**) DHA, and (**O**) 15d-PGJ2. The maximal response induced by 1 µM GW7647 (Figure 1A) was used as the 100% response. Data are expressed as the mean ± S.E. of 3 or 4 independent experiments with duplicate samples. The blue range was set to a single-digit range centered on the mean concentrations of human plasma samples previously reported in [24,25,26]. All raw data are available in Appendix A.

**Figure 3 ijms-26-12020-f003:**
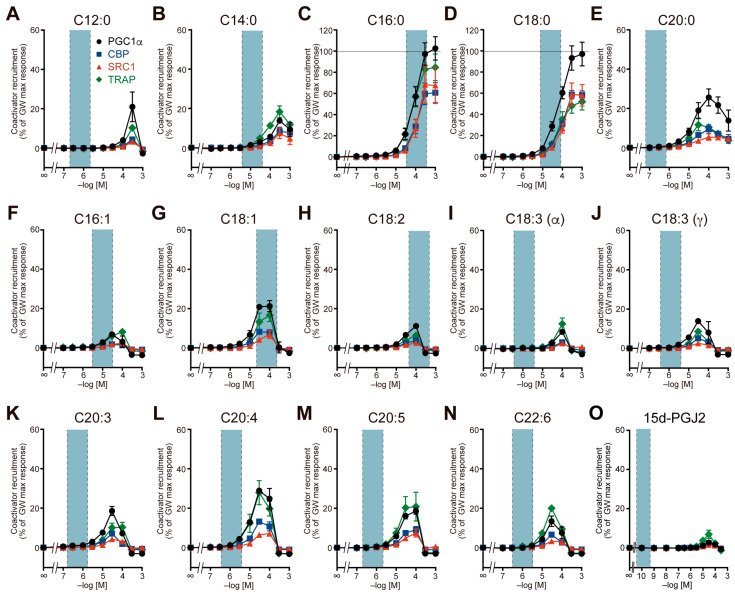
PPARδ activation by 15 endogenous fatty acids in a coactivator recruitment assay. Agonist-induced recruitment of four coactivator peptides (PGC1α in black circles; CBP in blue squares; SRC1 in red triangles; and TRAP in green diamonds) to human PPARδ-LBD was examined by using a cell-free TR-FRET assay. Concentration-dependent activity of (**A**) lauric acid, (**B**) myristic acid, (**C**) palmitic acid, (**D**) stearic acid, (**E**) arachidic acid, (**F**) palmitoleic acid, (**G**) oleic acid, (**H**) linoleic acid, (**I**) α-linolenic acid, (**J**) γ-linolenic acid, (**K**) dihomo-γ-linolenic acid, (**L**) arachidonic acid, (**M**) EPA, (**N**) DHA, and (**O**) 15d-PGJ2. The maximal response induced by 1 µM GW501516 (Figure 1B) was used as the 100% response. Data are expressed as the mean ± S.E. of 3 or 4 independent experiments with duplicate samples. The blue range was set to a single-digit range centered on the mean concentrations of human plasma samples previously reported in [24,25,26]. All raw data are available in Appendix A.

**Figure 4 ijms-26-12020-f004:**
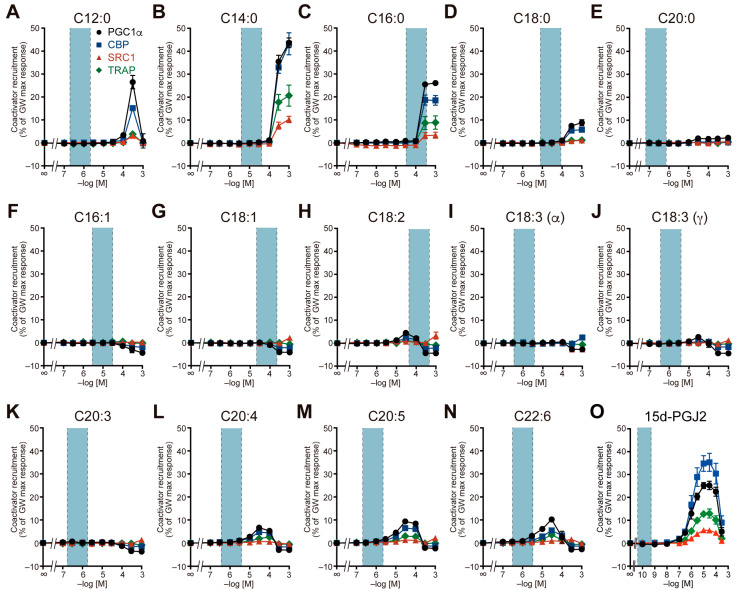
PPARγ activation by 15 endogenous fatty acids in a coactivator recruitment assay. Agonist-induced recruitment of four coactivator peptides (PGC1α in black circles; CBP in blue squares; SRC1 in red triangles; and TRAP in green diamonds) to human PPARγ-LBD was examined by using a cell-free TR-FRET assay. Concentration-dependent activity of (**A**) lauric acid, (**B**) myristic acid, (**C**) palmitic acid, (**D**) stearic acid, (**E**) arachidic acid, (**F**) palmitoleic acid, (**G**) oleic acid, (**H**) linoleic acid, (**I**) α-linolenic acid, (**J**) γ-linolenic acid, (**K**) dihomo-γ-linolenic acid, (**L**) arachidonic acid, (**M**) EPA, (**N**) DHA, and (**O**) 15d-PGJ2. The maximal response induced by 1 µM GW1929 (Figure 1C) was used as the 100% response. Data are expressed as the mean ± S.E. of three independent experiments with duplicate samples. The blue range was set to a single-digit range centered on the mean concentrations of human plasma samples previously reported in [24,25,26]. All raw data are available in Appendix A.

## Data Availability

The data used to support the findings of this study are available upon request from the corresponding author.

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
