# Peer review of "Int. J. Mol. Sci.2025, 26(24), 12020;https://doi.org/10.3390/ijms262412020"

_ijms, 2025, doi:10.3390/ijms262412020_

Round 1

Reviewer 1 Report

Comments and Suggestions for Authors

The manuscript investigates the activation profile of PPAR isoforms by endogenous long chain fatty acids (LCFA). The study evaluated the ability of 14 LCFA and PGJ2 in recruiting 4 different PPAR coactivators using the cell free TR-FRET assay.

The manuscript suffers several major critical points.

1) The experimental design bases on the observation that "Data on endogenous fatty acids ligands for PPARs are limited". This statement is inaccurate because several studies performed since the 1990s investigated the affinity of different fatty acids (saturated, MUFAs, PUFAs) to PPARs. For example, see Corton et al (doi: 10.1146/annurev.pharmtox.40.1.491.), Table 3.          For this reason, the scientific impact of the results is highly reduced.

2) The biological significance of the results obtained is unclear. In intracellular environment, the ability of LCFA-activated PPARs in recruiting different coactivators can be modulated by various factors/conditions not present in cell free experiments, including the relative LCFA availability and the relative expression of PPARs.

3) It is not correct to comment the ability of different LCFA in activating PPAR subtypes referring to the amount of LCFA in plasma considered as "physiological". It is well known that the amounts of LCFA is different in the cells and in plasma, the first being conditioned by the function performed by fatty acids the cell and the second mainly influenced by dietary uptake. Moreover, data regarding the content of LCFA in the plasma are reported in the section Results as if they had been obtained as part of this study, but they have been previously published.

4) Figure 1. The Figure describe the recruitment of coactivators in presence of GW7647 (PPARalpha), GW501516 (PPARdelta) and GW1929 (PPARgamma), pemafibrate, seladelpar, pioglitazone.

The Authors have previously published similar results (doi: 10.3390/biomedicines12030624) using the same experimental protocol, but the recruitment curves in relation to ligand concentration are significantly different. The authors must explain why they repeated the same evaluations and, above all, why the results are qualitatively different.

5) The Discussion section is very similar to the Results one. Moreover, the last sentence is highly speculative and not supported by the reported results.

6) in a similar way, the last sentences of the Abstract are not related to the results of the study

7) In the manuscript there are numerous incorrect statements.

a) line 38. Fibrates are no longer the first choice for reducing blood triglycerides and are only used when statins cannot be used.

b) lines 43-45. LDFA are not produced from certain types of eicosanoids.

c) lines 60-61. PGJ2 is one of the major chemical mediator of inflammation and is obvious that in basal conditions its level is low. Moreover, the references cited (8,9) are too old (2003) and do not correspond to current knowledge.

d) line 129. The authors affirm: "all 15 LCFA…", but LCFA examined are 14 since PGJ2 is nor a LCFA.

f) lines165-166. Arachidonic acid is not essential for human.

g) line 219. Reference 29 discusses on PPARalpha, not PPARgamma

Minor critical point

1) In Figure 1, panel A why does -FA- appear in the title?

2) line 200. Eliminate A before Althrough

Comments on the Quality of English Language

The English language requires minor editing revisions

Author Response

Reviewer #1

The manuscript investigates the activation profile of PPAR isoforms by endogenous long chain fatty acids (LCFA). The study evaluated the ability of 14 LCFA and PGJ2 in recruiting 4 different PPAR coactivators using the cell free TR-FRET assay. The manuscript suffers several major critical points.

We thank the Reviewer #1 for her/his time and helpful comments.

  • The experimental design bases on the observation that "Data on endogenous fatty acids ligands for PPARs are limited". This statement is inaccurate because several studies performed since the 1990s investigated the affinity of different fatty acids (saturated, MUFAs, PUFAs) to PPARs. For example, see Corton et al (doi: 10.1146/annurev.pharmtox.40.1.491.), Table 3. For this reason, the scientific impact of the results is highly reduced.

In Table 3 of the review by Corton et al. (2000), the PPARα/δ/γ activity of some free fatty acids in four older original studies (1992–1997) was rated on a scale from − to +++ without a specific rating scale. These were all based on cell-based reporter gene assays before the development of much higher sensitive TF-FRET-based coactivator recruitment assays, and only a single or a few very high concentrations between 1 and 100 µM were tested. As noted in our manuscript (new lines 51–55), there are some problems in principle (though there are some merits). The problem, we think, is that such knowledge about endogenous fatty acid ligands has not been significantly updated to date (see the latest Ref. 2 or other recent PPAR reviews such as 10.2217/fca-2016-0059; 10.2217/fca-2017-0019; 10.1155/2019/7242030), and this, we believe, is where the value of our experimental results lies. If not, we would like to know of any papers comparing the activation capacity of multiple endogenous fatty acids on the three PPAR subtypes using more sensitive assays, a wider dose range, and different coactivators.

  • The biological significance of the results obtained is unclear. In intracellular environment, the ability of LCFA-activated PPARs in recruiting different coactivators can be modulated by various factors/conditions not present in cell free experiments, including the relative LCFA availability and the relative expression of PPARs.

We believe that our results have biological significance in demonstrating that endogenous fatty acids and (clinically approved) synthetic PPAR subtype-selective agonists may recruit different coactivators, which thereby may regulate gene expression differently even via each of PPARα/δ/γ as evaluated by the Reviewer #2. Our results are intended to complement, but not replace, other experimental results, such as cell-based reporter gene assays, and suggest that in drug discovery, it may be necessary not only to focus on PPAR subtype selectivity, but also to investigate how they control coactivator recruitment and gene expression and whether this differs from the physiological phenomenon caused by endogenous fatty acids.

  • It is not correct to comment the ability of different LCFA in activating PPAR subtypes referring to the amount of LCFA in plasma considered as "physiological". It is well known that the amounts of LCFA is different in the cells and in plasma, the first being conditioned by the function performed by fatty acids the cell and the second mainly influenced by dietary uptake. Moreover, data regarding the content of LCFA in the plasma are reported in the section Results as if they had been obtained as part of this study, but they have been previously published.

Indeed, it may not be correct to consider blood concentrations of free fatty acids as “physiological.” However, it is still extremely difficult to accurately measure the concentration of free fatty acids in cells or nuclei. Therefore, we now refer to them as “physiologically relevant concentrations.” Furthermore, the legends of each figure more clearly indicate now that the free fatty acid concentrations shown as blue ranges in Figs. 2–4 are taken from other sources.

  • Figure 1. The Figure describe the recruitment of coactivators in presence of GW7647 (PPARalpha), GW501516 (PPARdelta) and GW1929 (PPARgamma), pemafibrate, seladelpar, pioglitazone. The Authors have previously published similar results (doi: 10.3390/biomedicines12030624) using the same experimental protocol, but the recruitment curves in relation to ligand concentration are significantly different. The authors must explain why they repeated the same evaluations and, above all, why the results are qualitatively different.

We use the GW series synthetic full agonists for quality control of each preparation of PPARα/δ/γ-LBD proteins. In this paper, we have improved the TR-FRET assay from the one used in 2024 Biomedicines. First, we used a Tecan Infinite 200 Pro F plex microplate reader equipped with appropriate filters, which offers >10 times the detection sensitivity compared to a Varioskan Flash Spectral Scan multimode reader equipped with variable filters used in all our previous papers. This change allowed us to reduce the use (and cost) of expensive Eu-W1024-labeled anti-6xHis antibody (from 8 nM to 4 nM) and ULight-streptavidin (from 80 nM to 40 nM). In addition, we deleted 1 mM dithiothreitol from PPAR-LBD mixture for the subsequent experiments (not this one) and used fatty acid-free (delipidized) PPARα-LBD in this study to accurately assess the effects of each added FA. GW1929 was obtained from Cayman Chemicals in this study but from Sigma–Aldrich in the previous study because the latter was out of stock. As a result, the activation patterns obtained were slightly different, but the reasons for this are unclear.

  • The Discussion section is very similar to the Results one. Moreover, the last sentence is highly speculative and not supported by the reported results.

The sentence “As for C16–C20, saturated LCFAs were more potent than unsaturated LCFAs, with all of the latter exhibiting bell-shaped responses (Figure 2F–2M) (original lines 178–180)” was deleted without disrupting the flow of the story. We have also deleted the last sentence.

  • In a similar way, the last sentences of the Abstract are not related to the results of the study.

The last two sentences were removed and the statement was toned down to “Such differences may be relevant to the pathogenesis of side effects of synthetic PPAR agonists occasionally observed in (pre)clinical studies (new lines 24–25).”

  • In the manuscript there are numerous incorrect statements.
  1. line 38. Fibrates are no longer the first choice for reducing blood triglycerides and are only used when statins cannot be used.

We stated as “Several PPAR agonists are currently used” but not “the (current) first choice.” The currently developed pemafibrate is also a TG-lowering fibrate. Here we just rephrased as “Several PPAR agonists were/are used” in new line 35.

  1. lines 43-45. LDFA are not produced from certain types of eicosanoids.

We were unsure first whether to classify 15d-PGJ2 as a certain type of LCFA, but if the reviewer thought that it was inappropriate, it should be excluded. Therefore, we deleted “from” here and modified relevant phrases throughout the manuscript.

  1. lines 60-61. PGJ2 is one of the major chemical mediators of inflammation and is obvious that in basal conditions its level is low. Moreover, the references cited (8,9) are too old (2003) and do not correspond to current knowledge.

Those papers were the first to dispute from the front that 15d-PDJ2 is an endogenous PPARγ ligand, and therefore we believe it is appropriate to cite them. Furthermore, as stated in the Discussion section, recent papers using highly sensitive MS quantification have shown that its levels in various diseases were not so particularly high. Of course, it is possible that extremely high concentrations can be achieved locally during severe inflammation, but this would not be considered a “physiological” action of 15d-PDJ2. We now stated in the last as “Clinical trials using multiple PPAR agonists are being conducted for autoimmune diseases, inflammatory diseases, infectious diseases, and malignant tumors [3]. Under these inflammatory conditions, the blood and local concentrations of many FAs are known to increase or even decrease, and it is possible that these endogenous FAs may be involved in the exacerbation or remission of these diseases via PPARs” (new lines 223–227).

  1. line 129. The authors affirm: "all 15 LCFA…", but LCFA examined are 14 since PGJ2 is nor a LCFA.

Please see our answer to 7-b).

  1. lines 165-166. Arachidonic acid is not essential for human.

We deleted “essential” here (new line 165).

  1. line 219. Reference 29 discusses on PPARalpha, not PPARgamma

We thank the reviewer for pointing out this error caused by the incorrect pickup from an Endnote list. Now corrected.

Minor critical point

  • In Figure 1, panel A why does –FA- appear in the title?

Different from our 2023 Biomedicines paper, we herein used delipidized PPARα-LBD. That is why we noted so. However, to avoid confusion, (–FA) was deleted.

2) line 200. Eliminate A before Although

We thank the reviewer. Now it was deleted.

Reviewer 2 Report

Comments and Suggestions for Authors

The important finding of the study is that different coactivator recruitment is the reason that synthetic PPAR agonists may not act on PPARs in the same way as endogenous fatty acids. 

This observation could be one reason that clinical development of many PPARγ-selective or PPAR dual/pan-agonists have been discontinued due to serious side effects, probably attributable to overstimulation of PPARγ.

Please mention also receptor independent activity profiles of synthetic PPARs. Toxicity of PPAR agonists may depend on PPAR dependent and independent mechanisms.

Please also shortly describe the experimental and clinical use of PPAR agonists in autoimmune disease, inflammation control and oncology, besides insulin resistance, in context with your experimental results as argument for promoting PPAR research.

PPARγ agonists may induce completely novel activity profiles in combination with imatinib, MEK inhibitors etc. by reprogramming non-oncogene addiction targets.

Author Response

Reviewer #2

The important finding of the study is that different coactivator recruitment is the reason that synthetic PPAR agonists may not act on PPARs in the same way as endogenous fatty acids. This observation could be one reason that clinical development of many PPARγ-selective or PPAR dual/pan-agonists have been discontinued due to serious side effects, probably attributable to overstimulation of PPARγ.

We thank the Reviewer #2 for her/his time and helpful comments.

  • Please mention also receptor independent activity profiles of synthetic PPARs. Toxicity of PPAR agonists may depend on PPAR dependent and independent mechanisms.

We thank the reviewer for this important input. We have now mentioned as “but some PPARγ agonists may have off-target effects, such as PPAR-independent genomic actions or non-genomic actions controlled by post-translational modifications [35–37] or may have entirely novel activities in combination with imatinib (a BCR-ABL tyrosine kinase inhibitor) or MEK inhibitors in cancer treatment [38–40]” in new lines 219–223.

  • Please also shortly describe the experimental and clinical use of PPAR agonists in autoimmune disease, inflammation control and oncology, besides insulin resistance, in context with your experimental results as argument for promoting PPAR research.

We have now stated in the last as “Clinical trials using multiple PPAR agonists are being conducted for autoimmune diseases, inflammatory diseases, infectious diseases, and malignant tumors [3]. Under these inflammatory conditions, the blood and local concentrations of many FAs are known to increase or even decrease, and it is possible that these endogenous FAs may be involved in the exacerbation or remission of these diseases via PPARs” (new lines 223–227).

  • PPARγ agonists may induce completely novel activity profiles in combination with imatinib, MEK inhibitors etc. by reprogramming non-oncogene addiction targets.

Please see our answer to 1).

Round 2

Reviewer 1 Report

Comments and Suggestions for Authors

The authors carried out some of the reviewer's suggestions, but with regards the major critical points they only replied in the response to the referee.

1)  With regard to the first comment of the referee, the arguments provided should be included in the manuscript, stating that knowledge of fatty acids as PPAR ligands/activators needs updating, but not that it is limited. For this reason, all the manuscript needs to be revised in the light of this observation.

2) Regarding the fact that the authors have previously published similar results with some activators (GW7647, GW501516, GW1929, pemafibrate, seladelpar, pioglitazone), they claimed to have done so because the methodology had been improved. This should be clearly stated in the manuscript to justify the repeating the experiments. 

However, no explanation for the observed difference in trends in the two manuscripts was provided and this remains a critical point.

Author Response

Reviewer #1

The authors carried out some of the reviewer's suggestions, but with regards the major critical points they only replied in the response to the referee.

We thank again the Reviewer #1 for her/his precious time and helpful comments. All changes are red-marked.

  • With regard to the first comment of the referee, the arguments provided should be included in the manuscript, stating that knowledge of fatty acids as PPAR ligands/activators needs updating, but not that it is limited. For this reason, all the manuscript needs to be revised in the light of this observation.

We (all non-native English speakers) apologize and regret that we were unable to recognize the subtle differences in nuance regarding English expressions. The first sentence “Data on endogenous fatty acid ligands for peroxisome proliferator-activated receptor α/δ/γ (PPARα/δ/γ) are limited” was revised by an English proofreading company. Now we are happy to replace it with “There is a wealth of information available about endogenous fatty acid ligands for peroxisome proliferator-activated receptor α/δ/γ (PPARα/δ/γ); however, there are limited examples of comparisons using standardized experimental systems (new lines 10–12).” We also modified as “There is a wealth of information available about the endogenous ligands of PPARs, which are presumably free long-chain fatty acids (LCFAs) produced prolifically via the lipolysis of dietary or stored triacylglycerols or certain types of eicosanoids derived from mem-brane phospholipids [2,7]. However, there are limited examples of comparisons using standardized experimental systems, and eicosanoids are produced in such small amounts that are difficult to quantify (or even identify), casting doubt on their involvement at physiological concentrations [8,9] (new lines 42–48).”

  • Regarding the fact that the authors have previously published similar results with some activators (GW7647, GW501516, GW1929, pemafibrate, seladelpar, pioglitazone), they claimed to have done so because the methodology had been improved. This should be clearly stated in the manuscript to justify the repeating the experiments.

We thank the reviewer for this nice advice and now added “As described later in the Material and Methods, we have improved the TR-FRET assay used in our previous paper [19] to be more sensitive, faster, and less expensive (new lines 81–83)” in the Results and “The improvements over the similar TR-FRET assay reported in our previous paper [19] were 1) the use of a new TECAN microplate reader equipped with appropriate filters, which offers 10 times more detection sensitivity and less than 1/10 the measurement time; 2) the reduction of the amounts of expensive Eu- and ULight-labelled reagents used by half; 3) the delipidation of PPARα-LBD (see above); and 4) the removal of 1 mM dithiothreitol from the assay mixture (not for this experiment, but for future experiments) (new lines 277–282)” in the Materials and Methods.

  • However, no explanation for the observed difference in trends in the two manuscripts was provided and this remains a critical point.

We agree, but when we tried it again with the old system, the results were the same. As for SRC1, it is the only peptide that contains Cys at the N-terminus of the peptide sequence, so we thought that there might be a difference depending on whether DTT was present or not, but the results were the same even when we experimented with a new SRC1 peptide from which only Cys was removed. However, what surprised us was that many endogenous fatty acids, unlike synthetic ligands, showed nearly identical coactivator recruitment responses to PPARα/δ/γ-LBD.

Round 3

Reviewer 1 Report

Comments and Suggestions for Authors

Changes made by the Authors correspond to the concept suggested by the referee and significantly improved the manuscript that is now suitable fro publication